# Metabolic Profiling Changes Induced by Fermented Blackberries in High-Fat-Diet-Fed Mice Utilizing Gas Chromatography–Mass Spectrometry Analysis

**DOI:** 10.3390/biology13070511

**Published:** 2024-07-09

**Authors:** Jae Young Park, Ha-Rim Kim, Seung-Hyeon Lee, Sang-Wang Lee, Hong-Sig Sin, Seon-Young Kim, Mi Hee Park

**Affiliations:** 1Jeonju AgroBio-Materials Institute, Wonjangdong-gil 111-27, Deokjin-gu, Jeonju-si 54810, Jeonbuk State, Republic of Korea; jjay1205@jami.re.kr (J.Y.P.); poshrim@jami.re.kr (H.-R.K.); sh94@jami.re.kr (S.-H.L.); 2Chebigen Inc., 62 Ballyong-ro, Deokjin-gu, Jeonju-si 54853, Jeonbuk State, Republic of Korea; cbg31@chebigen.com (S.-W.L.); shsdo@hanmail.net (H.-S.S.)

**Keywords:** blackberry, fermentation, metabolic profiling, GC-MS, obesity

## Abstract

**Simple Summary:**

This study identified a change in metabolites by fermented blackberries using GC-MS methods in high-fat-diet-induced mice. As a result, the altered metabolites in the high-fat-diet group consisted of 29 metabolites, showing 85% identity with known components in the database. Among the 29 metabolites, essential amino acids, non-essential amino acids, conditionally essential amino acids, and other metabolites were analyzed after treatment with blackberries and fermented blackberries in high-fat-diet-induced mice. This study constitutes a valuable resource on the dietary modification of fermented blackberries for obesity treatment approaches.

**Abstract:**

The aim of this study was to investigate the metabolic changes associated with the anti-obesity effects of fermented blackberry extracts in the liver tissues of high-fat-diet-fed mice using mass spectrometry-based metabolomics analysis. C57BL/6J mice were divided into eight groups: normal-diet-fed mice, high-fat-diet-fed mice, high-fat diet treated with blackberry extract, high-fat-diet mice treated with blackberry fermented by *L. plantarum*, and high-fat diet with blackberry fermented by *L. brevis*. After 12 weeks, the high-fat-diet group exhibited a greater increase in liver weight compared to the control group, and among the groups, the group administered with blackberry fermented with *L. plantarum* showed the most pronounced reduction in liver weight. As the primary organ responsible for amino acid metabolism, the liver is crucial for maintaining amino acid homeostasis. In our study, we observed that the levels of several essential amino acids, including isoleucine and valine, were decreased by the high-fat diet, and were recovered by administration of blackberry extract fermented with *L. plantarum*. Our results demonstrated the potential of blackberry extract fermented with *L. plantarum* as a functional material for metabolic disorders by restoring some of the amino acid metabolism disturbances induced by a high-fat diet.

## 1. Introduction

Metabolomics investigates the metabolites synthesized during biological transformations, elucidating metabolic pathways by analyzing the interconnected relationships among these molecules [1]. So, it has risen as a key asset in diagnosing and assessing disease and health conditions, enabling a detailed examination of small molecules within biological sources [2]. Key analytical techniques used in metabolomic research are mass spectrometry (MS) paired with gas chromatography (GC/MS), liquid chromatography (LC/MS), capillary electrophoresis (CE/MS), and nuclear magnetic resonance spectroscopy (NMR) [3]. Among these techniques, GC/MS stands out for its ability to effectively separate complex samples, its high sensitivity, and its resolution in identifying metabolites [4]. Metabolomic technology finds application in the development and assessment of functional foods due to its efficacy in monitoring environmental and dietary influences [5]. Through comprehensive metabolite profiling and multivariate statistical analysis, it offers valuable insights into the effects of these changes, aiding in the enhancement and evaluation of functional food products [6]. Recent advancements have seen the application of high-throughput metabolomic analyses to achieve deeper insight into the metabolic pathways and mechanisms implicated in metabolic diseases [7]. Metabolomics is assuming a progressively vital role in assessing changes in endogenous metabolites within biological fluids and tissues, revealing potential biomarkers for conditions like obesity and diabetes and laying the groundwork for new therapeutic advancements [8,9]. Metabolomic studies conducted on obese animal models have identified fatty acids, amino acids, carnitine, lysophosphatidylcholines, acyl-carnitines, and lysophosphatidylethanolamines as potential biomarkers indicative of obesity [10,11]. Using this information, researchers have conducted untargeted metabolite profiling on bioactive substances and specific chemical entities employed in the treatment of obesity [12].

Obesity arises from a disruption in the equilibrium of energy regulation, where the consumption of energy surpasses its expenditure [13]. Growing research indicates that alterations in dietary protein and its constituent amino acids influence energy equilibrium, leading to fluctuations in the mass of fat mass and overall body weight [14]. Nine indispensable amino acids include isoleucine (Ile), leucine (Leu), phenylalanine (Phe), methionine (Met), lysine (Lys), histidine (His), valine (Val), threonine (Thr), and tryptophan (Trp) [15]. Ile, Leu, and Val, collectively referred to as branched-chain amino acids, hold a distinct significance among essential amino acids [16]. Branched-chain amino acids have garnered significant interest in the past decade due to their capacity to stimulate protein synthesis and influence metabolism, accounting for approximately 40% of the body’s total amino acid demand [14]. Recent findings have conclusively demonstrated that essential amino acids function not solely as protein building blocks but also as pivotal signaling molecules that govern numerous biological processes [17,18]. The majority of studies have consistently shown that oral supplementation of leucine or isoleucine reduces fat mass and body weight in mice subjected to a high-fat diet (HFD) concurrently, yet not in mice already obese or rodents on a standard chow diet [19,20].

Blackberries, members of the Rubus genus, exhibit a diverse array of biological activities, including shielding against obesity, cancer, oxidative stress, age-related neurodegenerative disorders, endotoxicity, and cardiovascular diseases [21]. Additionally, blackberries boast proanthocyanidins and ellagitannins, both composed of ellagic acid polymers, which bolster their capacity to alleviate oxidative stress and inflammation [22]. Studies have indicated a substantial increase in phenolic compounds and gut metabolites during the digestion processes and fermentation of blackberries [21]. Fermented foods harbor a plethora of bioactive components, contributing to the prevention of arteriosclerosis, inflammation, and heart disease triggered through oxidative stress. Additionally, they are implicated in mitigating risks associated with diabetes, obesity, osteoporosis, and cancer [23]. During the fermentation of raw materials, numerous bioactive enzymes and chemicals are generated that are not inherent in the raw materials [24]. Fermented foods harbor enduring microorganisms, predominantly lactic acid bacteria, along with their principal metabolites [25]. Supplementation with both prebiotics and probiotics has shown beneficial effects on metabolic complications linked to obesity, including alterations in lipid profiles [26]. In our earlier research, we established that fermented blackberries exhibit safeguarding properties against skin aging induced by UV irradiation [27]. However, limited research has explored the effects of fermented blackberries on other diseases. Furthermore, our study demonstrated that fermented blackberries possess anti-obesity properties; so, we identify metabolites, including amino acids, in this study.

## 2. Materials and Methods

### 2.1. Preparation of Fermented Blackberries

*L. plantarum* JBMI F5 (KACC91638P) and *L. brevis* CBG-C24 (lactic acid bacteria isolated from vegetables) were grown in MRS broth (de Man, Rogosa and Sharpe) (Difco, Detroit, MI, USA) at 37 °C for 12–16 h. The raw blackberry material was then blended using a crusher. The crushed blackberry was filtered using a mesh to remove seeds. The fermentation process was conducted for 24 h while stirring at 150 rpm and 37 °C, using overnight cultured *L. plantarum* JBMI F5 or *L. brevis* CBG-C24. Following fermentation, the fermented blackberry was heated at 100 °C for 10 min and underwent lyophilization.

### 2.2. Animals

Four-week-old specific pathogen-free-grade male C57BL/6 mice were purchased from Damul Science (Daejeon, Republic of Korea) and were allowed to acclimate for 1 week. The mice were kept in mouse cage under a 12 h light/dark cycle at 22 °C ± 2 °C and a relative humidity of 55% ± 5%. All experimental procedures were approved by the Animal Care Committee of Jeonju AgroBio-Materials Institute, Jeonju, Korea (approval number: JAMI IACUC 2023006).

### 2.3. Experimental Groups

The mice were divided into eight groups: normal group (N), the high-fat-diet-induced obesity group (HFD), the blackberry treated group (BB 100 and BB 500 mg/kg), and the fermented blackberry administration group (FBB-LP 100 mg/kg, FBB-LP 500 mg/kg, FBB-LB 100 mg/kg, and FBB-LB 500 mg/kg). To induce obesity, the HFD, BB, FBB-LP, and FBB-LB groups were fed a diet consisting of 60% kcal from fat for 12 weeks, while the N group was fed a normal diet containing 10% kcal from fat. The BB, FBB-LP, and FBB-LB groups were orally administered BBs (100 and 500 mg/kg), FBB-LP (100 and 500 mg/kg), and FBB-LB (100 mg/kg and 500 mg/kg), respectively, for 12 weeks. N and HFD groups were administered vehicle (distilled water). Five mice per group were analyzed.

### 2.4. Sample Preparation and Derivatization

Mouse liver tissue, 20 ± 1 mg, was placed in 700 μL mixture of cold MeOH/CHCl3 (3:1, *v/v*), followed by homogenization using a Mixer Mill (MM400, Retsch Technology, Haan, Germany). The samples were vortexed for 1 min and centrifuged at 13,000 rpm for 15 min at 4 °C. Then, 100 μL of supernatant was moved to a 2 mL vial and dried in a SpeedVac concentration at room temperature before derivatization. Dried samples were treated with 100 μL BSTFA and incubated for 30 min at 45 °C. After cooling, the internal standard of methyl salicylate at 10 ppm was added, followed by vortexing, before GC-MS analysis.

### 2.5. GC-MS Conditions

For GC-MS analysis, a Shimadzu GCMS-QP2020 NX (Shimadzu, Kyoto, Japan) in-strument was utilized, equipped with a DB-5MS fused silica capillary column (30 m, 0.25 mm, 0.25 μm, J&W Scientific, Folsom, CA, USA). The carrier gas was high-purity helium, and the injec-tion mode was split (10:1) with an injection volume of 1 μL. The temperatures of the inter-face and ion source were set to 280 °C, while the injection port temperature was 250 °C. The GC oven gradient temperatures were adjusted as follows: the initial oven temperature was held at 60 °C for 2 min, raised with a rate of 10 °C/min to 300 °C, and maintained for 10 min. The electron ionization voltage was set to 70 eV, and the *m*/*z* range was 50 to 600 for analysis conducted in full scan mode.

### 2.6. Statistical Analysis

Data are presented as means ± standard deviation. Statistical analyses were conducted using Sigmaplot v16.0 (Systat Software Inc., San Jose, CA, USA). Statistical analysis was performed to identify differences, followed by one-way analysis of variance and Turkey’s multiple comparison test. These analyses were employed to assess differences among three or more groups for all measured parameters. Statistical significance was defined as a difference with a *p*-value of less than 0.05. For the metabolomic analysis, GC-MS data (metabolite peak intensity) were analyzed using MetaboAnalyst v6.0 (http://www.metaboanalyst.ca/) accessed on 12 March 2024. Multivariate analysis, including principal component analysis (PCA) and partial least squares discriminant analysis (PLS-DA), was conducted using the MultiQuant software in MetaboAnalyst 6.0. PLS-DA was performed with variable importance in projection (VIP) analysis. Compound identification was conducted using the NIST20 databases (NIST/EPA/NIH Mass Spectral Library).

## 3. Results

### 3.1. Effects of Fermented Blackberries Administration on Liver Weights in HFD-Fed Mice

We showed the HFD-induced obesity models using C57BL/6 mice which are suitable for extrapolation to human studies [28]. demonstrated a significant increase in liver weight in the HFD group compared to the N group. The results showed that after 12 weeks, the liver weight of the HFD group increased to 1.38 ± 0.14 g, which is about 1.3 times that of the N group, while the fermented blackberries administration group had a relatively low increase in liver weight (Figure 1). We demonstrated that administration of fermented blackberries by *L. plantarum* (FBB-LP) significantly reduced the liver weight at the concentrations of 100 and 500 mg/kg, whereas the liver weight was reduced by treatment of fermented blackberries by *L. Brevis* (FBB-LB) at a concentration of 500 mg/kg. Therefore, it can be inferred that fermented blackberries, especially FBB-LP, hold therapeutic potential for combating HFD-induced obesity.

### 3.2. Differential Metabolomic Analysis between Groups in HFD-Fed Mice

GC-MS analysis revealed various endogenous metabolites in the liver tissues of mice in the N and HFD groups, of which 29 metabolites exhibited 85% similarity with identified components (Figure 2 and Table 1). PCA and PLS-DA analyses demonstrated clear separation between the data of the experimental groups (Figure 3), highlighting significant differences in the metabolic profiles of the liver tissue among the experimental groups.

### 3.3. Effect of Fermented Blackberries on Essential Amino Acid in the Liver Tissue of HFD-Fed Mice

Among the 29 metabolites from GC-MS analysis, we analyzed the essential amino acids in the experimental groups (Figure 4). We showed that the isoleucine and valine were significantly reduced by a high-fat diet and reversed by administration of 500 mg/kg of blackberries fermented by *L. plantarum* JBMI F5. However, threonine and phenylalanine were significantly reduced by a high-fat diet, whereas they were not changed by treatment with fermented blackberries.

### 3.4. Effect of Fermented Blackberries on Non-Essential Amino Acid in the Liver Tissue of HFD-Fed Mice

Among the 29 metabolites from GC-MS analysis, we also analyzed the non-essential amino acids in the experimental groups (Figure 5). We showed that the alanine, glutamic acid, and serine were significantly reduced by a high-fat diet, whereas they were not significantly changed by treatment with fermented blackberries.

### 3.5. Effect of Fermented Blackberries on Conditionally Essential Amino Acids in the Liver Tissue of HFD-Fed Mice

Among the 29 metabolites from GC-MS analysis, we also analyzed the conditionally essential amino acids in the experimental groups (Figure 6). We showed that the level of glycine was significantly reduced by a high-fat diet and reversed by administration of fermented blackberries. Interestingly, the level of proline was significant in both blackberries and fermented blackberries. The level of tyrosine was significantly reduced by a high-fat diet, whereas it was not significantly changed by treatment with fermented blackberries.

### 3.6. Effect of Blackberries and Fermented Blackberries on Glucose and Mannose Metabolisms in the Liver Tissue of HFD-Fed Mice

Among the 29 metabolites from GC-MS analysis, we also analyzed the glucose and mannose metabolisms in the experimental groups. We showed that the level of lactic acid, the final product of glucose metabolism, was significantly enhanced by a high-fat diet and reduced by administration of blackberries and fermented blackberries (Figure 7A). β-D-glucopyranose was induced by a high-fat diet; however, it was not significantly reduced in any of the sample treatment groups (Figure 7B). Moreover, the levels of β-L-Mannofuranose (Figure 7C) and a-D-Mannopyranose (Figure 7D) were significantly enhanced by a high-fat diet and decreased by administration of blackberries and fermented blackberries.

## 4. Discussion

In this study, we demonstrated that metabolites, especially amino acids, are significantly changed by fermented blackberries in an HFD-induced obese mice model, correlated with an increase in liver weight. Our results revealed that the essential amino acids, especially Ile and Val, were dramatically increased by treatment with the FBB-LP 500 mg/kg group, indicating that the *L. plantarum* JBMI F5 was related to the production of Ile and Val. We showed that threonine and phenylalanine were significantly reduced by a high-fat diet, whereas they were not changed by treatment with fermented blackberries. We also analyzed the non-essential amino acids in the experimental groups. We showed that alanine, glutamic acid, and serine were significantly reduced by a high-fat diet, whereas they were not significantly changed by treatment with fermented blackberries. We showed that the level of the conditionally essential amino acid glycine was significantly reduced by a high-fat diet and reversed by administration of fermented blackberries. Interestingly, the level of proline was significantly increased by administration of both blackberries and fermented blackberries. The level of tyrosine was significantly reduced by a high-fat diet, whereas it was not significantly changed by treatment with fermented blackberries. Moreover, lactic acid, the final product of glucose metabolism, and levels of β-L-Mannofuranose and α-D-Mannopyranose were significantly increased by a high-fat diet and decreased by administration of blackberries and fermented blackberries. Interestingly, we found that the glucose metabolite lactic acid was significantly reduced by both blackberries and fermented blackberries. Moreover, the mannose metabolites Mannofuranose and Mannopyranose were also reduced by both blackberries and fermented blackberries. Although further research and mechanism research need to be conducted, it is believed that these metabolites are all related to the blackberry itself. Additionally, these are predicted to be associated with products that do not change during fermentation.

Obesity is frequently attributed to an imbalance between energy intake, which is often excessive, and energy expenditure, which may be insufficient [29]. Several factors include hormones, neurotransmitters, and metabolic pathways that regulate energy expenditure, food intake, and the storage and utilization of nutrients [30]. Indeed, the metabolic aspects discussed typically pertain to adult mammals, considering the relative significance of essential amino acids can vary significantly for growing mammals due to their heightened requirements for growth and development [31]. Amino acids are categorized into essential and non-essential types. Essential amino acids are crucial dietary components, as the body cannot synthesize them in adequate amounts. Dietary proteins serve as the primary sources of essential amino acids [32]. It has been widely speculated among researchers that diet plays a pivotal role in the onset of metabolic disorders and obesity [33]. There is a scarcity of studies that comprehensively assess the metabonomic changes in the primary tissues of animal models. Furthermore, there is a lack of data comparing the impacts of high-fat diets on the metabolism of critical tissues [34]. Thus, our objective was to thoroughly assess the influence of diet on the metabolomic changes in major tissues of animal models. The experimental setup of this study involved comparing the metabolomic profiles of the major tissues in mice treated with fermented blackberries following a high-fat diet, utilizing a GC/MS-based metabolomics approach. As a result, this study provides insights into the metabolic changes that occur in mice within an obesogenic environment. This research also aimed to explore optimal nutritional requirements and feeding regimes that may attenuate the effects of a high-fat diet.

Amino acids can be categorized as glucogenic, ketogenic, or both, depending on the metabolic products they produce during catabolism [35]. The amino acid pool is sufficiently abundant to fulfill the requirements of protein synthesis. Additionally, surplus amino acids are metabolized into compounds that can enter the TCA cycle to generate energy in the form of ATP, thereby contributing approximately 10% to the body’s overall energy supply [18]. Amino acids that do not directly enter the TCA cycle are categorized as either ketogenic, contributing to the formation of fatty acids and ketone bodies, or glucogenic, contributing to the formation of glucose [18]. Ketogenic amino acids are metabolized into acetyl-CoA or acetoacetate. Glucogenic amino acids are metabolized into pyruvate, fumarate, succinyl-CoA, α-ketoglutarate, or oxaloacetate. Amino acids that exhibit both ketogenic and glucogenic properties include phenylalanine, tryptophan, isoleucine, tyrosine, and threonine [33]. Lysine and leucine are exclusively ketogenic, while the remaining amino acids are purely glucogenic: aspartate, asparagine, alanine, glutamine, arginine, glutamate, histidine, proline, valine, methionine, serine, cysteine, and glycine [36]. Several clinical trials and animal studies have demonstrated that supplementation with ketogenic amino acids can have a positive impact on insulin sensitivity and/or obesity [37]. For example, feeding mice leucine has been shown to reduce high-fat-diet-induced hyperglycemia, obesity, and hypercholesterolemia [38], and an orally administered mixture of ketogenic amino acids including leucine, isoleucine, valine, threonine, and lysine has been demonstrated to improve insulin sensitivity in elderly patients with type-2 diabetes [39]. Additionally, in knockout mice lacking the mitochondrial branched-chain amino acid transaminase, increased availability of branched-chain amino acids preserved muscle insulin sensitivity, even under conditions of long-term high-fat feeding [40]. Therefore, the role of essential amino acids, particularly ketogenic amino acids, in the progression of insulin resistance and hepatic steatosis remains a topic of controversy. Furthermore, previous studies have demonstrated that dietary phenolic compounds can exert a protective effect on the liver by modulating lipid metabolism and enhancing antioxidant capacity [41,42]. This implies that specific dietary components may positively influence metabolic disorders induced by a high-fat diet through similar mechanisms.

Our research findings offer convincing proof of the advantageous effects of administering fermented blackberries on metabolic disorders. Notably, the administration of fermented blackberries led to a notable decrease in liver weight induced by a high-fat diet, indicating its potential as an effective intervention for managing metabolic disorders. Furthermore, our study demonstrated promising outcomes for metabolic health, showing that administration of fermented blackberries was linked to the production of essential amino acids. Studies indicate that regular consumption of fermented foods may play a role in weight management and in preventing metabolic disorders associated with obesity [23]. In this study, we found that fermented blackberries induced essential amino acid levels, indicating its potential in dietary functional foods.

In summary, our findings robustly endorse the concept that fermented blackberries exert a preventive effect against metabolic disorders. Further investigation is needed to clarify the underlying mechanisms and maximize the therapeutic potential of fermented blackberries.

## 5. Conclusions

This research provides compelling scientific evidence supporting the impact of fermented blackberries in mice fed a high-fat diet, focusing on the change in metabolites. This study revealed that fermented blackberries significantly reduced liver weight gain induced by a high-fat diet. We observed a tendency to restore the levels of several essential amino acids, including isoleucine and valine, that were reduced by the high-fat diet in the group administered blackberry extract fermented with *L. plantarum*. Our results demonstrated the potential of blackberry extract fermented with *L. plantarum* as a functional material for metabolic disorders by restoring essential amino acid disturbances induced by a high-fat diet.

## Figures and Tables

**Figure 1 biology-13-00511-f001:**
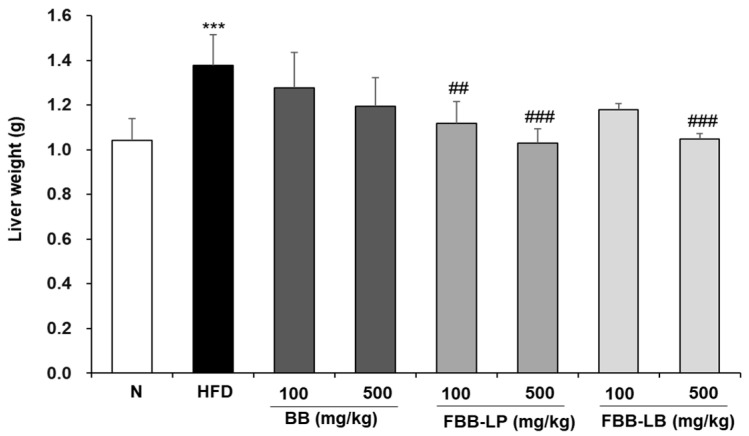
Effects of fermented blackberries on liver weight in HFD-induced obesity mice. Liver weights 12 weeks after initiating obesity with HFD. All values are presented as mean ± SD. Data were analyzed using Turkey’s multiple comparison test. *** *p* < 0.001 versus the N group; ^##^
*p* < 0.01 and ^###^
*p* < 0.001, versus the HFD group; N, normal; HFD, high-fat diet; BB, blackberry; FBB-LP, blackberries fermented by *L. plantarum* JBMI F5; FBB-LB, blackberries fermented by *L. brevis* CBG-C24.

**Figure 2 biology-13-00511-f002:**
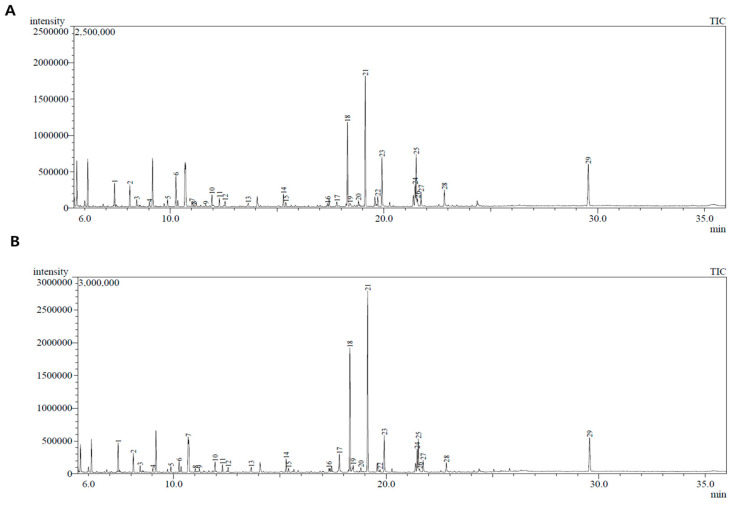
GC-MS analysis of liver tissues. (**A**) GC-MS analysis of liver tissues of N group. (**B**) GC-MS analysis of liver tissues of HFD group.

**Figure 3 biology-13-00511-f003:**
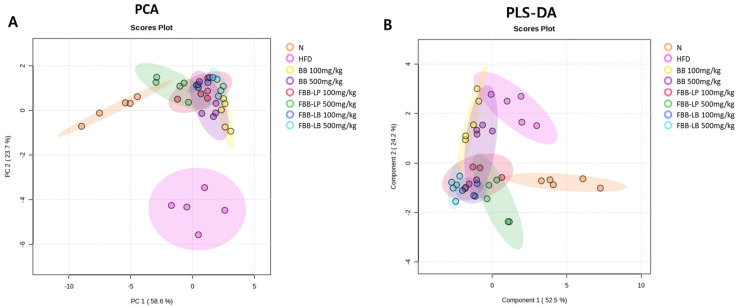
PCA (**A**) and PLS-DA (**B**) plots of the metabolome of mouse liver tissues. N, normal; HFD, high-fat diet; BB, blackberry; FBB-LP, blackberries fermented by *L. plantarum* JBMI F5; FBB-LB, blackberries fermented by *L. brevis* CBG-C24.

**Figure 4 biology-13-00511-f004:**
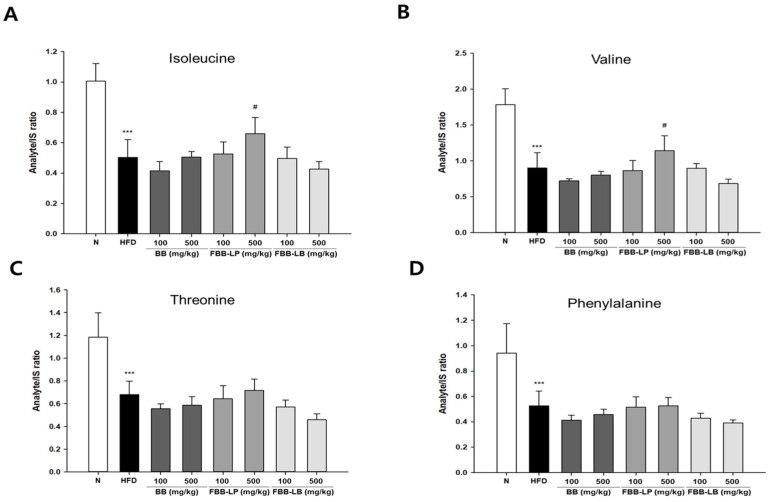
GC-MS analysis of essential amino acids in liver tissues of HFD-induced mice. (**A**) Isoleucine, (**B**) Valine, (**C**) Threonine and (**D**) Phenylalanine were analyzed. All values are expressed as mean ± SD. Data were analyzed using Turkey’s multiple comparison test. *** *p* < 0.001 versus the N group; ^#^
*p* < 0.05 versus the HFD group; N, normal; HFD, high-fat diet; BB, blackberry; FBB-LP, blackberries fermented by *L. plantarum* JBMI F5; FBB-LB, blackberries fermented by *L. brevis* CBG-C24.

**Figure 5 biology-13-00511-f005:**
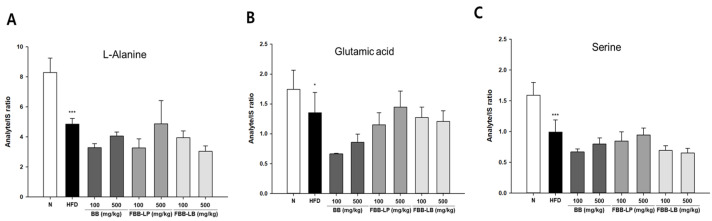
GC-MS analysis of non-essential amino acids in liver tissues of HFD-induced mice. (**A**) L-Alanine, (**B**) Glutamic acid and (**C**) Serine were analyzed. All values are expressed as mean ± SD. Data were analyzed using Turkey’s multiple comparison test. * *p* < 0.05, and *** *p* < 0.001 versus the N group; N, normal; HFD, high-fat diet; BB, blackberry; FBB-LP, blackberries fermented by *L. plantarum* JBMI F5; FBB-LB, blackberries fermented by *L. brevis* CBG-C24.

**Figure 6 biology-13-00511-f006:**
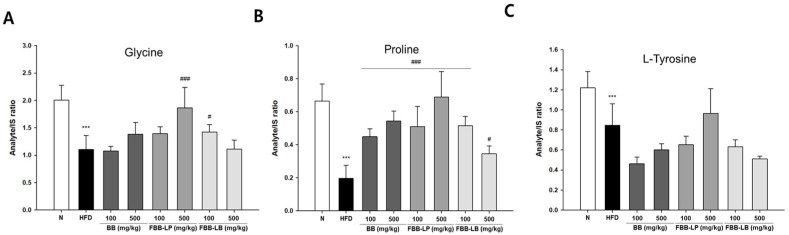
GC-MS analysis of conditionally essential amino acids in liver tissues of HFD-induced mice. (**A**) Glycine, (**B**) Proline and (**C**) L-Tyrosine were analyzed. All values are expressed as mean ± SD. Data were analyzed using Turkey’s multiple comparison test. *** *p* < 0.001 versus the N group; ^#^
*p* < 0.05 and ^###^
*p* < 0.001, versus the HFD group; N, normal; HFD, high-fat diet; BB, blackberry; FBB-LP, blackberries fermented by *L. plantarum* JBMI F5; FBB-LB, blackberries fermented by *L. brevis* CBG-C24.

**Figure 7 biology-13-00511-f007:**
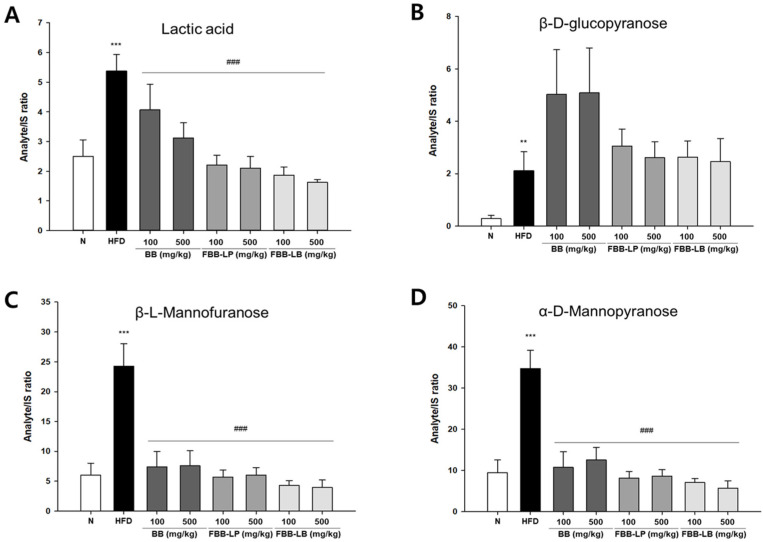
GC-MS analysis of metabolites in liver tissues of HFD-induced mice. (**A**) Lactic acid, (**B**) β-D-glucopyranose, (**C**) β-L-Mannofuranose and (**D**) α-D-Mannopyranose were analyzed. All values are expressed as mean ± SD. Data were analyzed using Turkey’s multiple comparison test. ** *p* < 0.01 and *** *p* < 0.001 versus the N group; ^###^
*p* < 0.001 versus the HFD group; N, normal; HFD, high-fat diet; BB, blackberry; FBB-LP, blackberries fermented by *L. plantarum* JBMI F5; FBB-LB, blackberries fermented by *L. brevis* CBG-C24.

**Table 1 biology-13-00511-t001:** Compound name in GC-MS analysis.

ID	Compound Name
1	Lactic acid
2	L-Alanine
3	Glycine
4	3-Hydroxybutyric acid
5	L-Valine
6	Urea
7	L-Isoleucine
8	L-Proline
9	2-Butenedioic acid
10	Serine
11	L-Threonine
12	Methyl salicylate
13	Malic acid
14	L-Glutamic acid
15	Phenylalanine
16	Glucofuranoside
17	Beta-D-glucopyranose
18	Beta-L-Mannofuranose
19	D-Galactose
20	L-Tyrosine
21	Alpha-D-Mannopyranose
22	Palmitelaidic acid
23	Palmitic Acid
24	9,12-Octadecadienoic acid
25	9-Octadecenoic acid
26	Oleic acid
27	Stearic acid
28	Arachidonic acid
29	Cholesterol

## Data Availability

The data presented in this study are available in this article.

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
