# Peer review of "Metabolic Profiling Changes Induced by Fermented Blackberries in High-Fat-Diet-Fed Mice Utilizing Gas Chromatography–Mass Spectrometry Analysis"

_biology, 2024, doi:10.3390/biology13070511_

Round 1
Reviewer 1 Report
Comments and Suggestions for Authors
The manuscript investigated the metabolic changes associated with the anti-obesity effects of fermented blackberry in the liver tissues of high-fat diet-fed mice using mass spectrometry-based metabolomics analysis. The study designed well and obtained some interesting information. The following are my comments, which need to be addressed before it can be accepted.
The title requires revision as it does not reflect the main animal subjects of this paper. The study pertains to mice; however, the current wording of the title could misleadingly suggest "Fermented Black-berries" as the focus. A more appropriate title should clearly indicate the subject of study to avoid confusion. A possible revised title could be:
"Metabolic Profiling Changes Induced by Fermented Blackberries in High-Fat Diet-Fed Mice utilizing Gas Chromatography-Mass Spectrometry Analysis
Lines 66-68, the sentence “Amino acids serve not only as the fundamental ………. that govern various biological pathways [15].” is redundant with the following sentence(lines 75-77)“that essential amino acids function not solely as protein building blocks but also as pivotal signaling molecules that govern numerous biological processes [18]”. It is better to delete one of them.
Line 104, “MRS (de Man, Rogosa and 104 Sharpe) broth (Difco, usa)”, it should be “MRS broth (de Man, Rogosa and Sharpe, Difco, USA)
Line 105, “Blackberry” should be “blackberry”
Lines 117-124, “The mice were divided into five groups”, However, in this study, the mice were divided into eight groups; please make the correction. Additionally, please specify the number of mice in each group.
Line 127,“added” should be deleted.
Line 161, “Figure 1A” should be “Figure 1”.
Line 157-166, this study evaluates the effects of fermented blackberries based on liver weight. However, how do different treatments affect the liver index of mice (the ratio of liver weight to body weight)? Generally, organ indices are more suitable for assessing the physical condition or metabolic status of animals.。
Line 176, “NC” should be “N”. The designation for the control group should be consistent throughout the text.
Figure 4, Figure 5 and Figure 6 were not cited in the text.
In the figure 4 legends, it is necessary to specify the p-value represented by the symbol '#' in the figure. The phrases "##P < 0.01, and ###P < 0.001" should be removed, as these symbols (##, ###) do not appear in the figure.
Lines 208 and 261, “Alanine” should be “alanine”.
In the figure 5 legends, it is necessary to specify the p-value represented by the symbol * in the figure. The phrase "##P < 0.01, and ###P < 0.001" should be removed, as these symbols (##, ###) do not appear in the figure.
In the figure 6 legends, it is necessary to specify the p-value represented by the symbol '#' in the figure. The phrases “##P < 0.01, and” should be removed, as the symbol ## does not appear in the figure.
In the figure 7 legends, the phrases “##P < 0.01, and” should be removed, as the symbol ## does not appear in the figure.
Line 265, add the word “increased” after “the level of proline was significantly”.
In the "Discussion" section (lines 255-289), where the authors discussed the impact of fermented blackberries on liver lipid metabolism, the following citation could be incorporated:
Zhang, L.; Zhang, J.; Zang, H.; Yin, Z.; Guan, P.; Yu, C.; Shan, A.; Feng, X. Dietary pterostilbene exerts potential protective effects by regulating lipid metabolism and enhancing antioxidant capacity on liver in broilers. Journal of Animal Physiology and Animal Nutrition 2024, 108, 1-13. https://doi.org/10.1111/jpn.13941.
Guan, P.; Yu, H.; Wang, S.; Sun, J.; Chai, X.; Sun, X.; Qi, X.; Zhang, R.; Jiao, Y.; Li, Z.; Kim, I.; Feng, X.; Liu, X. Dietary rutin alleviated the damage by cold stress on inflammation reaction, tight junction protein and intestinal microbial flora in the mice intestine. The Journal of Nutritional Biochemistry 2024, 130, 109658. https://doi.org/10.1016/j.jnutbio.2024.109658
"Furthermore, the previous studies demonstrated that dietary phenolic compounds can exert a protective effect on the liver by modulating lipid metabolism and enhancing antioxidant capacity (Zhang et al., 2024; Guan et al., 2024). This implies that specific dietary components may positively influence metabolic disorders induced by a high-fat diet through similar mechanisms"
These citations can provide readers with additional background information on the potential role of dietary components in modulating metabolism and preventing obesity-related diseases.
Line 289-290, “Therefore, in order to evaluate the role of diet in metabolomic variations of the main tissues in animal models comprehensively.” it is not a complete sentence, please rewrite it.
Comments on the Quality of English LanguageModerate editing of English language required
Reviewer 2 Report
Comments and Suggestions for Authors
Line 176 change “HDF” to “HFD”.
In the study, the HFD consist of..? how much is the total calories that have been administered to the mouse? How much the calories for carbohydrate, protein and fats?
In materials and methods please add the source of the LAB strains. Whether the strains come from ATCC or isolated from food sources
What is the marker for obesity other than amino acids content in the liver? Since in the study only reveal the amino acids and fatty acids.
Why the authors use these strains of LP and LB?
What is the correlation of the observed amino acids with obesity? Does this AA have thermogenic effect?
It is interesting to see other mechanisms that play crucial role as antiobesity in fermented blackberries, whether the LAB (several LAB possess antiobesity), the blackberry also contain flavonoid which may contribute for antiobesity, or significant increase in SOD/GPX/NOx in the mouse liver that may increase fat oxidation.
Why the authors did not orally administered LP and LB without BB?
The mice were orally administered with a range of 100 and 500 mg/kg BW. What is the reason using such dose?
What is the positive control and negative control for the diet experiment?
In the Discussion, please rewrite the first paragraph because it’s already written in the Results. The Discussion must be improved.
In the Discussion please also add that some AA essentials and non-essentials act as glucogenic AA and ketogenic AA may contribute to obesity effect.
Round 2
Reviewer 1 Report
Comments and Suggestions for Authors
line 22: "five groups" should be changed to "eight groups"
Comments on the Quality of English LanguageMinor editing of English language required
Author Response
Point: line 22: "five groups" should be changed to "eight groups"
Response: Thank you for your valuable comment. We corrected.
Reviewer 2 Report
Comments and Suggestions for Authors
Dear Editor,
It seems that the similarity is quite high 44% and it would be nice if the authors could reduce it to 20%, thank you
Author Response
Thank you for your valuable comment. We revised the manuscript.